# Online Adaptive Methods, Universality and Acceleration

**Kfir Y. Levy**
ETH Zurich
yehuda.levy@inf.ethz.ch

**Alp Yurtsever**
EPFL
alp.yurtsever@epfl.ch

**Volkan Cevher**
EPFL
volkan.cevher@epfl.ch

## Abstract

We present a novel method for convex unconstrained optimization that, *without any modifications*, ensures: **(i)** accelerated convergence rate for smooth objectives, **(ii)** standard convergence rate in the general (non-smooth) setting, and **(iii)** standard convergence rate in the stochastic optimization setting. To the best of our knowledge, this is the first method that *simultaneously* applies to all of the above settings.

At the heart of our method is an adaptive learning rate rule that employs importance weights, in the spirit of adaptive online learning algorithms [12, 20], combined with an update that linearly couples two sequences, in the spirit of [2]. An empirical examination of our method demonstrates its applicability to the above mentioned scenarios and corroborates our theoretical findings.

## 1 Introduction

The accelerated gradient method of Nesterov [23] is one of the cornerstones of modern optimization. Due to its appeal as a computationally efficient and fast method, it has found use in numerous applications including: imaging [8], compressed sensing [14], and deep learning [31], amongst other.

Despite these merits, accelerated methods are less prevalent in Machine Learning due to two major issues: **(i)** acceleration is inappropriate for handling noisy feedback, and **(ii)** acceleration requires the knowledge of the objective's smoothness. While each of these issues was separately resolved in [17, 16, 33], and respectively in [25]; it was unknown whether there exists an accelerated method that addresses both issues. In this work we propose such a method.

Concretely, Nesterov [25] devises a method that obtains an accelerated convergence rate of $\mathcal{O}(1/T^2)$ for smooth convex objectives, and a standard rate of $\mathcal{O}(1/\sqrt{T})$ for non-smooth convex objectives, over $T$ iterations. This is done without any prior knowledge of the smoothness parameter, and is therefore referred to as a *universal*[1] method. Nonetheless, this method uses a line search technique in every round, and is therefore inappropriate for handling noisy feedback. On the other hand, Lan [17], Hu et al. [16], and Xiao [33], devise accelerated methods that are able to handle noisy feedback and obtain a convergence rate of $\mathcal{O}(1/T^2 + \sigma/\sqrt{T})$, where $\sigma$ is the variance of the gradients. However, these methods are not universal since they require the knowledge of both $\sigma$ and of the smoothness.

Conversely, adaptive first order methods are very popular in Machine Learning, with AdaGrad, [12], being the most prominent method among this class. AdaGrad is an online learning algorithm which adapts its learning rate using the feedback (gradients) received through the optimization process, and is known to successfully handle noisy feedback. This renders AdaGrad as

the method of choice in various learning applications. Note however, that AdaGrad (probably) can not ensure acceleration. Moreover, it was so far unknown whether AdaGrad is at all able to exploit smoothness in order to converge faster.

In this work we investigate unconstrained convex optimization. We suggest AcceleGrad (Alg. 2), a novel *universal* method which employs an accelerated-gradient-like update rule together with an adaptive learning rate à la AdaGrad. Our contributions,

- We show that AcceleGrad obtains an accelerated rate of $\mathcal{O}(1/T^2)$ in the smooth case and $\tilde{\mathcal{O}}(1/\sqrt{T})$ in the general case, without any prior information of the objective's smoothness.

- We show that *without any modifications*, AcceleGrad ensures a convergence rate of $\tilde{\mathcal{O}}(1/\sqrt{T})$ in the general stochastic convex case.

- We also present a new result regarding the AdaGrad algorithm. We show that in the case of stochastic optimization with a smooth expected loss, AdaGrad ensures an $\mathcal{O}(1/T + \sigma/\sqrt{T})$ convergence rate, where $\sigma$ is the variance of the gradients. AdaGrad does not require a knowledge of the smoothness, hence this result establishes the universality of AdaGrad (though without acceleration).

On the technical side our algorithm emoploys three simultaneous mechanisms: learning rate adaptation in conjunction with importance weighting, in the spirit of adaptive online learning algorithms [12, 20], combined with an update rule that linearly couples two sequences, in the spirit of [2].

This paper is organized as follows. In Section 2 we present our setup and review relevant background. Our results and analysis for the offline setting are presented in Section 3, and Section 4 presents our results for the stochastic setting. In Section 5 we present our empirical study, and Section 6 concludes.

**Related Work:** In his pioneering work, Nesterov [23], establishes an accelerated rate for smooth convex optimization. This was later generalized in, [24, 6], to allow for general metrics and line search.

In recent years there has been a renewed interest in accelerated methods, with efforts being made to understand acceleration as well as to extend it beyond the standard offline optimization setting.

An extension of acceleration to handle stochastic feedback was developed in, [17, 16, 33, 9]. Acceleration for modern variance reduction optimization methods is explored in, [29, 1], and generic templates to accelerating variance reduction algorithms are developed in, [21, 15]. Scieur et al. [28], derives a scheme that enables hindsight acceleration of non-accelerated methods. In [34], the authors devise a universal accelerated method for primal dual problems. And the connection between acceleration and ODEs is investigated in, [30, 32, 13, 19, 5, 4]. Universal accelerated schemes are explore in [25, 18, 26], yet these works do not apply to the stochastic setting. Alternative accelerated methods and interpretations are explored in, [3, 7, 11].

Curiously, Allen-Zhu and Orecchia [2], interpret acceleration as a linear coupling between gradient descent and mirror descent, our work builds on their ideas. Our method also relies on ideas from [20], where universal (non-accelerated) procedures are derived through a conversion scheme of online learning algorithms.

## 2 Setting and Preliminaries

We discuss the optimization of a convex function $f : \mathbb{R}^d \mapsto \mathbb{R}$. Our goal is to (approximately) solve the following unconstrained optimization problem,

$$\min_{x \in \mathbb{R}^d} f(x) \ .$$

We focus on first order methods, i.e., methods that only require gradient information, and consider both smooth and non-smooth objectives. The former is defined below,

**Definition 1** ($\beta$-smoothness)**.** *A function $f : \mathbb{R}^d \mapsto \mathbb{R}$ is $\beta$-smooth if,*

$$f(y) \leq f(x) + \nabla f(x) \cdot (y - x) + \frac{\beta}{2} \|x - y\|^2; \quad \forall x, y \in \mathbb{R}^d$$

---
**Algorithm 1** Adaptive Gradient Method (AdaGrad)
---
**Input**: #Iterations $T$, $x_1 \in \mathcal{K}$, set $\mathcal{K}$
**for** $t = 1 \dots T$ **do**

    Calculate: $g_t = \nabla f(x_t)$, and update, $\eta_t = D \left( 2 \sum_{\tau=1}^{t} \|g_\tau\|^2 \right)^{-1/2}$
    Update:
$$x_{t+1} = \Pi_\mathcal{K} \left( x_t - \eta_t g_t \right)$$

**end for**
Output: $\bar{x}_T = \frac{1}{T} \sum_{t=1}^{T} x_t$

---

It is well known that with the knowledge of the smoothness parameter, $\beta$, one may obtain fast convergence rates by an appropriate adaptation of the update rule. In this work we do not assume any such knowledge; instead we assume to be given a bound on the distance between some initial point, $x_0$, and a global minimizer of the objective.

This is formalized as follows: we are given a compact convex set $\mathcal{K}$ that contains a global minimum of $f$, i.e., $\exists z \in \mathcal{K}$ such that $z \in \arg\min_{x \in \mathbb{R}^d} f(x)$. Thus, for any initial point, $x_0 \in \mathcal{K}$, its distance from the global optimum is bounded by the diameter of the set, $D := \max_{x,y \in \mathcal{K}} \|x - y\|$. Note that we allow to choose points outside $\mathcal{K}$. We also assume that the objective $f$ is $G$-Lipschitz, which translates to a bound of $G$ on the magnitudes of the (sub)-gradients.

An access to the exact gradients of the objective is not always possible. And in many scenarios we may only access an oracle which provides noisy and unbiased gradient estimates. This *Stochatic Optimization* setting is prevalent in Machine Learning, and we discuss it more formally in Section 4.

**The AdaGrad Algorithm:** The adaptive method presented in this paper is inspired by AdaGrad (Alg. 1), a well known online optimization method which employs an adaptive learning rate. The following theorem states AdaGrad's guarantees[2] , [12],

**Theorem 2.1.** *Let $\mathcal{K}$ be a convex set with diameter $D$. Let $f$ be a convex function. Then Algorithm 1 guarantees the following error;*

$$f(\bar{x}_T) - \min_{x \in \mathcal{K}} f(x) \le \sqrt{2D^2 \sum_{t=1}^{T} \|g_t\|^2 / T} \ .$$

**Notation:** We denote the Euclidean norm by $\| \cdot \|$. Given a compact convex set $\mathcal{K}$ we denote by $\Pi_\mathcal{K}(\cdot)$ the projection onto the $\mathcal{K}$, i.e. $\forall x \in \mathbb{R}^d$, $\Pi_\mathcal{K}(x) = \arg\min_{y \in \mathcal{K}} \|y - x\|^2$ .

## 3  Offline Setting

This section discusses the offline optimization setting where we have an access to the exact gradients of the objective. We present our method in Algorithm 2, and substantiate its universality by providing $O(1/T^2)$ rate in the smooth case (Thm. 3.1), and a rate of $O(\sqrt{\log T / T})$ in the general convex case (Thm. 3.2). The analysis for the smooth case appears in Section 3.1 and we defer the proof of the non-smooth case to the Appendix.

AcceleGrad is summarized in Algorithm 2. Inspired by, [2], our method linearly couples between two sequences $\{z_t\}_t, \{y_t\}_t$ into a sequence $\{x_{t+1}\}_t$. Using the gradient , $g_t = \nabla f(x_{t+1})$, these sequences are then updated with the same learning rate, $\eta_t$, yet with different *reference points* and *gradient magnitudes*. Concretely, $y_{t+1}$ takes a gradient step starting at $x_{t+1}$. Conversely, for $z_{t+1}$ we scale the gradient by a factor of $\alpha_t$ and then take a projected gradient step starting at $z_t$. Our method finally outputs a weighted average of the $\{y_{t+1}\}_t$ sequence.

Our algorithm coincides with the method of [2] upon taking $\eta_t = 1/\beta$ and outputting the last iterate, $y_T$, rather then a weighted average; yet this method is not

**Algorithm 2** Accelerated Adaptive Gradient Method (AcceleGrad)

---

**Input**: #Iterations $T$, $x_0 \in \mathcal{K}$, diameter $D$, weights $\{\alpha_t\}_{t \in [T]}$, learning rate $\{\eta_t\}_{t \in [T]}$
Set: $y_0 = z_0 = x_0$
**for** $t = 0 \ldots T$ **do**
   Set $\tau_t = 1/\alpha_t$
   Update:

$$x_{t+1} = \tau_t z_t + (1 - \tau_t) y_t , \quad \text{and define} \;\; g_t := \nabla f(x_{t+1})$$
$$z_{t+1} = \Pi_{\mathcal{K}} \left( z_t - \alpha_t \eta_t g_t \right)$$
$$y_{t+1} = x_{t+1} - \eta_t g_t$$

**end for**
Output: $\bar{y}_T \propto \sum_{t=0}^{T-1} \alpha_t y_{t+1}$

---

universal. Below we present our $\beta$-*independent* choice of learning rate and weights,

$$\eta_t = \frac{2D}{\left( G^2 + \sum_{\tau=0}^{t} \alpha_\tau^2 \|g_\tau\|^2 \right)^{1/2}} \qquad \& \qquad \alpha_t = \begin{cases} 1 & 0 \le t \le 2 \\ \frac{1}{4}(t+1) & t \ge 3 \end{cases} \tag{1}$$

The learning rate that we suggest adapts similarly to AdaGrad. Differently from AdaGrad we consider the importance weights, $\alpha_t$, inside the learning rate rule; an idea that we borrow from [20]. The weights that we employ are increasing with $t$, which in turn emphasizes recent queries.

Next we state the guarantees of AcceleGrad for the smooth and non-smooth cases,

**Theorem 3.1.** *Assume that $f$ is convex and $\beta$-smooth. Let $\mathcal{K}$ be a convex set with bounded diameter $D$, and assume there exists a global minimizer for $f$ in $\mathcal{K}$. Then Algorithm 2 with weights and learning rate as in Equation* (1) *ensures,*

$$f(\bar{y}_T) - \min_{x \in \mathbb{R}^d} f(x) \le \mathcal{O}\left( \frac{DG + \beta D^2 \log(\beta D/G)}{T^2} \right)$$

**Remark:** Actually, in the smooth case we do not need a bound on the Lipschitz continuity, i.e., $G$ is only required in case that the objective is non-smooth. Concretely, if we know that $f$ is smooth then we may use $\eta_t = 2D\left( \sum_{\tau=0}^{t} \alpha_\tau^2 \|g_\tau\|^2 \right)^{-1/2}$, which yields a rate of $\mathcal{O}\left( \frac{\beta D^2 \log(\beta D/\|g_0\|)}{T^2} \right)$.

Next we show that the exactly same algorithm provides guarantees in the general convex case,

**Theorem 3.2.** *Assume that $f$ is convex and $G$-Lipschitz. Let $\mathcal{K}$ be a convex set with bounded diameter $D$, and assume there exists a global minimizer for $f$ in $\mathcal{K}$. Then Algorithm 2 with weights and learning rate as in Equation* (1) *ensures,*

$$f(\bar{y}_T) - \min_{x \in \mathbb{R}^d} f(x) \le \mathcal{O}\left( GD\sqrt{\log T}/\sqrt{T} \right)$$

**Remark:** For non-smooth objectives, we can modify AcceleGrad and provide guarantees for the *constrained* setting. Concretely, using Alg. 2 with a projection step for the $y_t$'s, i.e., $y_{t+1} = \Pi_{\mathcal{K}}(x_{t+1} - \eta_t g_t)$, then we can bound its error by $f(\bar{y}_T) - \min_{x \in \mathcal{K}} f(x) \le \mathcal{O}\left( GD\sqrt{\log T}/\sqrt{T} \right)$. This holds even in the case where minimizer over $\mathcal{K}$ is not a global one.

### 3.1 Analysis of the Smooth Case

Here we provide a proof sketch for Theorem 3.1. For brevity, we will use $z \in \mathcal{K}$ to denote a *global mimimizer* of $f$ which belongs to $\mathcal{K}$.

Recall that Algorithm 2 outputs a weighted average of the queries. Consequently, we may employ Jensen's inequality to bound its error as follow,

$$f(\bar{y}_T) - f(z) \le \frac{1}{\sum_{t=0}^{T-1} \alpha_t} \sum_{t=0}^{T-1} \alpha_t \left( f(y_{t+1}) - f(z) \right) . \tag{2}$$

Combining this with $\sum_{t=0}^{T-1} \alpha_t \geq \Omega(T^2)$, implies that in order to substantiate the proof it is sufficient to show that, $\sum_{t=0}^{T-1} \alpha_t \left(f(y_{t+1}) - f(z)\right)$, is bounded by a constant. This is the bulk of the analysis.

We start with the following lemma which provides us with a bound on $\alpha_t \left(f(y_{t+1}) - f(z)\right)$,

**Lemma 3.1.** *Assume that $f$ is convex and $\beta$-smooth. Then for any sequence of non-negative weights $\{\alpha_t\}_{t \geq 0}$, and learning rates $\{\eta_t\}_{t \geq 0}$, Algorithm 2 ensures the following to hold,*

$$\alpha_t(f(y_{t+1}) - f(z)) \leq (\alpha_t^2 - \alpha_t)(f(y_t) - f(y_{t+1})) + \frac{\alpha_t^2}{2}\left(\beta - \frac{1}{\eta_t}\right)\|y_{t+1} - x_{t+1}\|^2$$

$$+ \frac{1}{2\eta_t}\left(\|z_t - z\|^2 - \|z_{t+1} - z\|^2\right)$$

Interestingly, choosing $\eta_t \leq 1/\beta$, implies that the above term, $\frac{\alpha_t^2}{2}\left(\beta - \frac{1}{\eta_t}\right)\|y_{t+1} - x_{t+1}\|^2$, does not contribute to the sum. We can show that this choice facilitates a concise analysis establishing an error of $\mathcal{O}(\beta D^2/T^2)$ for $\bar{y}_T$[3].

Note however that our learning rate does not depend on $\beta$, and therefore the mentioned term is not necessarily negative. This issue is one of the main challenges in our analysis. Next we provide a proof sketch of Theorem 3.1. The full proof is deferred to the Appendix.

*Proof Sketch of Theorem 3.1.* Lemma 3.1 enables to decompose $\sum_{t=0}^{T-1} \alpha_t(f(y_{t+1}) - f(z))$,

$$\sum_{t=0}^{T-1} \alpha_t(f(y_{t+1}) - f(z)) \leq \underbrace{\sum_{t=0}^{T-1} \frac{1}{2\eta_t}\left(\|z_t - z\|^2 - \|z_{t+1} - z\|^2\right)}_{(A)}$$

$$+ \underbrace{\sum_{t=0}^{T-1}(\alpha_t^2 - \alpha_t)(f(y_t) - f(y_{t+1}))}_{(B)} + \underbrace{\sum_{t=0}^{T-1} \frac{\alpha_t^2}{2}\left(\beta - \frac{1}{\eta_t}\right)\|y_{t+1} - x_{t+1}\|^2}_{(C)}$$

$$(3)$$

Next we separately bound each of the above terms.

**(a) Bounding (A) :** Using the fact that $\{1/\eta_t\}_{t \in [T]}$ is monotonically increasing allows to show,

$$\sum_{t=0}^{T-1} \frac{1}{2\eta_t}\left(\|z_t - z\|^2 - \|z_{t+1} - z\|^2\right) \leq \frac{1}{2}\sum_{t=1}^{T-1} \|z_t - z\|^2\left(\frac{1}{\eta_t} - \frac{1}{\eta_{t-1}}\right) + \frac{\|z_0 - z\|^2}{2\eta_0} \leq \frac{D^2}{2\eta_{T-1}}$$

$$(4)$$

where we used $\|z_t - z\| \leq D$.

**(b) Bounding (B) :** We will require the following property of the weights that we choose (Eq. (1)),
$$(\alpha_t^2 - \alpha_t) - (\alpha_{t-1}^2 - \alpha_{t-1}) \leq \alpha_{t-1}/2 \tag{5}$$

Now recall that $z := \arg\min_{x \in \mathbb{R}^d} f(x)$, and let us denote the sub-optimality of $y_t$ by $\delta_t$, i.e. $\delta_t = f(y_t) - f(z)$. Noting that $\delta_t \geq 0$ we may show the following,

$$\sum_{t=0}^{T-1}(\alpha_t^2 - \alpha_t)\left(f(y_t) - f(y_{t+1})\right) = \sum_{t=0}^{T-1}(\alpha_t^2 - \alpha_t)\left(\delta_t - \delta_{t+1}\right)$$

$$\leq \sum_{t=1}^{T-1}((\alpha_t^2 - \alpha_t) - (\alpha_{t-1}^2 - \alpha_{t-1}))\delta_t$$

$$\leq \frac{1}{2}\sum_{t=0}^{T-1} \alpha_t\left(f(y_{t+1}) - f(z)\right) \tag{6}$$

Where the last inequality uses Equation (5) (see full proof for the complete derivation).

**(c) Bounding** $(C)$ **:**   Let us denote $\tau_\star := \max\left\{t \in \{0, \ldots, T-1\} : 2\beta \geq 1/\eta_t\right\}$ . We may now split the term $(C)$ according to $\tau_\star$,

$$
(C) = \sum_{t=0}^{\tau_\star} \frac{\alpha_t^2}{2}\left(\beta - \frac{1}{\eta_t}\right)\|y_{t+1} - x_{t+1}\|^2 + \sum_{t=\tau_\star+1}^{T-1} \frac{\alpha_t^2}{2}\left(\beta - \frac{1}{\eta_t}\right)\|y_{t+1} - x_{t+1}\|^2
$$

$$
\leq \frac{\beta}{2}\sum_{t=0}^{\tau_\star}\alpha_t^2\|y_{t+1} - x_{t+1}\|^2 - \frac{1}{4}\sum_{t=\tau_\star+1}^{T-1}\frac{\alpha_t^2}{\eta_t}\|y_{t+1} - x_{t+1}\|^2
$$

$$
= \frac{\beta}{2}\sum_{t=0}^{\tau_\star}\eta_t^2\alpha_t^2\|g_t\|^2 - \frac{1}{4}\sum_{t=\tau_\star+1}^{T-1}\eta_t\alpha_t^2\|g_t\|^2 \tag{7}
$$

where in the second line we use $2\beta \leq \frac{1}{\eta_t}$ which holds for $t > \tau_\star$, implying that $\beta - \frac{1}{\eta_t} \leq -\frac{1}{2\eta_t}$; in the last line we use $\|y_{t+1} - x_{t+1}\| = \eta_t\|g_t\|$.

**Final Bound :**   Combining the bounds in Equations (4),(6),(7) into Eq. (3), and re-arranging gives,

$$
\frac{1}{2}\sum_{t=0}^{T-1}\alpha_t(f(y_{t+1}) - f(z)) \leq \underbrace{\frac{D^2}{2\eta_{T-1}} - \frac{1}{4}\sum_{t=\tau_\star+1}^{T-1}\eta_t\alpha_t^2\|g_t\|^2}_{(*)} + \underbrace{\frac{\beta}{2}\sum_{t=0}^{\tau_\star}\eta_t^2\alpha_t^2\|g_t\|^2}_{(**)} \tag{8}
$$

We are now in the intricate part of the proof where we need to show that the above is bounded by a constant. As we show next this crucially depends on our choice of the learning rate. To simplify the proof sketch we assume to be using , $\eta_t = 2D\left(\sum_{\tau=0}^{t}\alpha_\tau^2\|g_\tau\|^2\right)^{-1/2}$, i.e. taking $G = 0$ in the learning rate. We will require the following lemma before we go on,

**Lemma.** *For any non-negative numbers $a_1, \ldots, a_n$ the following holds:*

$$
\sqrt{\sum_{i=1}^{n}a_i} \leq \sum_{i=1}^{n}\frac{a_i}{\sqrt{\sum_{j=1}^{i}a_j}} \leq 2\sqrt{\sum_{i=1}^{n}a_i} \ .
$$

Equipped with the above lemma and using $\eta_t$ explicitly enables to bound $(*)$,

$$
(*) = \frac{D}{4}\left(\sum_{t=0}^{T-1}\alpha_t^2\|g_t\|^2\right)^{1/2} - \frac{D}{2}\sum_{t=\tau_\star+1}^{T-1}\frac{\alpha_t^2\|g_t\|^2}{\left(\sum_{\tau=0}^{t}\alpha_\tau^2\|g_\tau\|^2\right)^{1/2}}
$$

$$
\leq \frac{D}{4}\sum_{t=0}^{T-1}\frac{\alpha_t^2\|g_t\|^2}{\left(\sum_{\tau=0}^{t}\alpha_\tau^2\|g_\tau\|^2\right)^{1/2}} - \frac{D}{2}\sum_{t=\tau_\star+1}^{T-1}\frac{\alpha_t^2\|g_t\|^2}{\left(\sum_{\tau=0}^{t}\alpha_\tau^2\|g_\tau\|^2\right)^{1/2}}
$$

$$
\leq \frac{D}{4}\sum_{t=0}^{\tau_\star}\frac{\alpha_t^2\|g_t\|^2}{\left(\sum_{\tau=0}^{t}\alpha_\tau^2\|g_\tau\|^2\right)^{1/2}}
$$

$$
\leq \frac{D}{2}\left(\sum_{\tau=0}^{\tau_\star}\alpha_\tau^2\|g_\tau\|^2\right)^{1/2}
$$

$$
= \frac{D^2}{\eta_{\tau_\star}} \leq 2\beta D^2
$$

where in the last inequality we have used the definition of $\tau_\star$ which implies that $1/\eta_{\tau_\star} \leq 2\beta$.

Using similar argumentation allows to bound the term $(**)$ by $\mathcal{O}(\beta D^2 \log(\beta D/\|g_0\|))$. Plugging these bounds back into Eq. (8) we get,

$$
\sum_{t=0}^{T-1}\alpha_t(f(y_{t+1}) - f(z)) \leq \mathcal{O}(\beta D^2 \log(\beta D/\|g_0\|)) \ .
$$

Combining this with Eq. (2) and noting that $\sum_{t=0}^{T-1}\alpha_t \geq T^2/32$, concludes the proof.   $\square$

# 4 Stochastic Setting

This section discusses the stochastic optimization setup which is prevalent in Machine Learning scenarios. We formally describe this setup and prove that Algorithm 2, *without any modification*, is ensured to converge in this setting (Thm. 4.1). Conversely, the universal gradient methods presented in [25] rely on a line search procedure, which requires exact gradients and function values, and are therefore inappropriate for stochastic optimization.

As a related result we show that the AdaGrad algorithm (Alg. 1) is universal and is able to exploit small variance in order to ensure fast rates in the case of stochastic optimization with smooth expected loss (Thm. 4.2). We emphasize that AdaGrad does not require the smoothness nor a bound on the variance. Conversely, previous works with this type of guarantees, [33, 17], require the knowledge of *both of these parameters*.

**Setup:** We consider the problem of minimizing a convex function $f : \mathbb{R}^d \mapsto \mathbb{R}$. We assume that optimization lasts for $T$ rounds; on each round $t = 1, \ldots, T$, we may query a point $x_t \in \mathbb{R}^d$, and receive a *feedback*. After the last round, we choose $\bar{x}_T \in \mathbb{R}^d$, and our performance measure is the expected excess loss, defined as,

$$\mathbf{E}[f(\bar{x}_T)] - \min_{x \in \mathbb{R}^d} f(x) \ .$$

Here we assume that our feedback is a first order noisy oracle such that upon querying this oracle with a point $x$, we receive a bounded and unbiased gradient estimate, $\tilde{g}$, such

$$\mathbf{E}[\tilde{g}|x] = \nabla f(x); \quad \& \quad \|\tilde{g}\| \leq G \tag{9}$$

We also assume that the internal coin tosses (randomizations) of the oracle are independent. It is well known that variants of Stochastic Gradient Descent (SGD) are ensured to output an estimate $\bar{x}_T$ such that the excess loss is bounded by $O(1/\sqrt{T})$ for the setups of stochastic convex optimization, [22]. Similarly to the offline setting we assume to be given a set $\mathcal{K}$ with bounded diameter $D$, such that there exists a global optimum of $f$ in $\mathcal{K}$.

The next theorem substantiates the guarantees of Algorithm 2 in the stochastic case,

**Theorem 4.1.** *Assume that $f$ is convex and $G$-Lipschitz. Let $\mathcal{K}$ be a convex set with bounded diameter $D$, and assume there exists a global minimizer for $f$ in $\mathcal{K}$. Assume that we invoke Algorithm 2 but provide it with noisy gradient estimates (see Eq. (9)) rather then the exact ones. Then Algorithm 2 with weights and learning rate as in Equation (1) ensures,*

$$\mathbf{E}[f(\bar{y}_T)] - \min_{x \in \mathbb{R}^d} f(x) \leq \mathcal{O}\left(GD\sqrt{\log T}/\sqrt{T}\right)$$

The analysis of Theorem 4.1 goes along similar lines to the proof of its offline counterpart (Thm. 3.2).

It is well known that AdaGrad (Alg. 1) enjoys the standard rate of $\mathcal{O}(GD/\sqrt{T})$ in the stochastic setting. The next lemma demonstrates that: **(i)** AdaGrad is universal, and **(ii)** AdaGrad implicitly make use of smoothness and small variance in the stochastic setting.

**Theorem 4.2.** *Assume that $f$ is convex and $\beta$-smooth. Let $\mathcal{K}$ be a convex set with bounded diameter $D$, and assume there exists a global minimizer for $f$ in $\mathcal{K}$. Assume that we invoke AdaGrad (Alg. 1) but provide it with noisy gradient estimates (see Eq. (9)) rather then the exact ones. Then,*

$$\mathbf{E}[f(\bar{x}_T)] - \min_{x \in \mathbb{R}^d} f(x) \leq \mathcal{O}\left(\frac{\beta D^2}{T} + \frac{\sigma D}{\sqrt{T}}\right)$$

*where $\sigma^2$ is a bound on the variance of noisy gradients, i.e., $\forall x \in \mathbb{R}^d$; $\mathbf{E}\left[\|\tilde{g} - \nabla f(x)\|^2|x\right] \leq \sigma^2$ .*

# 5 Experiments

In this section we compare AcceleGrad against AdaGrad (Alg. 1) and universal gradient methods [25], focusing on the effect of tuning parameters and the level of adaptivity.

We consider smooth ($p = 2$) and non-smooth ($p = 1$) regression problems of the form

$$\min_{x \in \mathbb{R}^d} F(x) := \|Ax - b\|_p^p \ .$$

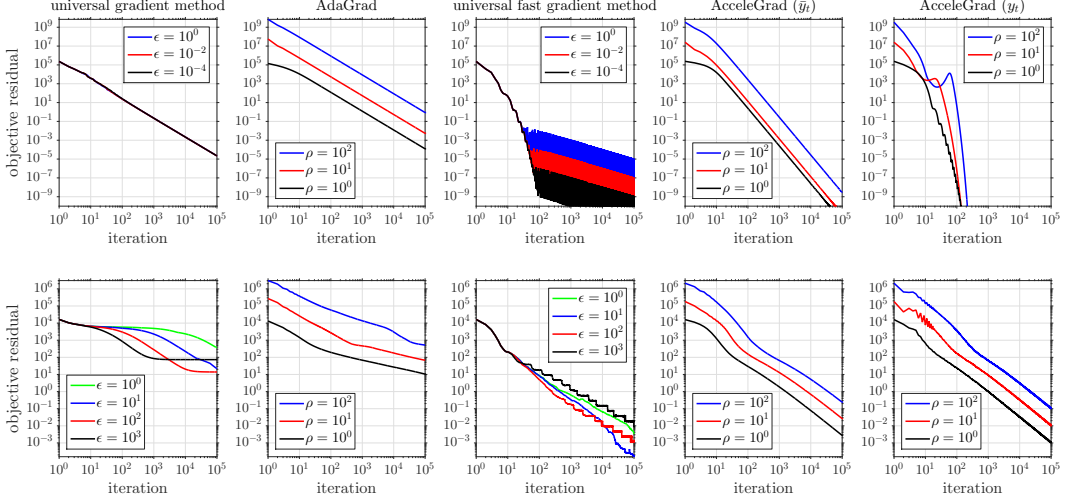

Figure 1: Comparison of universal methods at a smooth *(top)* and a non-smooth *(bottom)* problem.

We synthetically generate matrix $A \in \mathbb{R}^{n \times d}$ and a point of interest $x^\natural \in \mathbb{R}^d$ randomly, with entries independently drawn from standard Gaussian distribution. Then, we generate $b = Ax^\natural + \omega$, with Gaussian noise, $w \sim \mathcal{N}(0, \sigma^2)$ and $\sigma^2 = 10^{-2}$. We fix $n = 2000$ and $d = 500$.

Figure 1 presents the results for the offline optimization setting, where we provide the exact gradients of $F$. All methods are initialized at the origin, and we choose $\mathcal{K}$ as the $\ell_2$ norm ball of diameter $D$.

Universal gradient methods are based on an inexact line-search technique that requires an input parameter $\epsilon$. Moreover, these methods have convergence guarantees only up to $\frac{\epsilon}{2}$-suboptimality. For smooth problems, these methods perform better with smaller $\epsilon$. In stark contrast, for the non-smooth problems, small $\epsilon$ causes late adaptation, and large $\epsilon$ ends up with early saturation. Tuning is a major problem for these methods, since it requires rough knowledge of the optimal value.

Universal gradient method (also the fast version) provably requires two line-search iterations on average at each outer iteration. Consequently, it performs two data pass at each iteration (four for the fast version), while AdaGrad and AcceleGrad require only a single data pass.

The parameter $\rho$ denotes the ratio between $D/2$ and the distance between initial point and the solution. Parameter $D$ plays a major role on the step-size of AdaGrad and AcceleGrad. Overestimating $D$ causes an overshoot in the first iterations. AcceleGrad consistently overperforms AdaGrad in the deterministic setting. As a final note, it needs to be mentioned that the iterates $y_t$ of AcceleGrad empirically converge faster than the averaged sequence $\bar{y}_T$. Note that for AcceleGrad we always take $G = 0$, i.e., use $\eta_t = 2D \left( \sum_{\tau=0}^t \alpha_\tau^2 \|g_\tau\|^2 \right)^{-1/2}$.

We also study the stochastic setup (see Appendix), where we provide noisy gradients of $F$ based on minibatches. As expected, universal line search methods *fail* in this case, while AcceleGrad converges and performs similarly to AdaGrad.

**Large batches:** In the appendix we show results on a real dataset which demonstrate the appeal of AcceleGrad in the large-minibatch regime. We show that with the increase of batch size the performance of AcceleGrad verses the number of gradient calculations does not degrade and might even *improve*. This is beneficial when we like to parallelize a stochastic optimization problem. Conversely, for AdaGrad we see a clear degradation of the performance as we increase the batch size.

# 6    Conclusion and Future Work

We have presented a novel universal method that may exploit smoothness in order to accelerate while still being able to successfully handle noisy feedback. Our current analysis only applies to unconstrained optimization problems. Extending our work to the constrained setting is a natural

future direction. Another direction is to implicitly adapt the parameter $D$, this might be possible using ideas in the spirit of scale-free online algorithms, [27, 10].

**Acknowledgments**

The authors would like to thank Zalán Borsos for his insightful comments on the manuscript.

This project has received funding from the European Research Council (ERC) under the European Union's Horizon 2020 research and innovation programme (grant agreement no 725594 - time-data). K.Y.L. is supported by the ETH Zurich Postdoctoral Fellowship and Marie Curie Actions for People COFUND program.

## Footnotes

[1]Following Nesterov's paper [25], we say that an algorithm is *universal* if it does not require to know in advance whether the objective is smooth or not. Note that universality does not mean a parameter free algorithm. Specifically, Nesterov's universal methods [25] as well as ours are not parameter free.

[2]Actually AdaGrad is well known to ensure regret guarantees in the online setting. For concreteness, Thm. 2.1 provides error guarantees in the offline setting.

[3]While we do not spell out this analysis, it is a simplified version of our proof for Thm. 3.1.

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
