[Supplementary Material]

# Online Adaptive Methods, Universality and Acceleration
**(Full version with supplementary material)**

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

# A  Proofs for the Smooth Case (Thm. 3.1)

Here we provide the complete proof of Theorem 3.1, and of the related lemmas. For brevity, we will use $z \in \mathcal{K}$ to denote a *global mimimizer* of $f$ which belongs to $\mathcal{K}$.

Recall that Algorithm 2 outputs a weighted average of the queries. Consequently, we may employ Jensen's inequality to bound its error as follow,

$$f(\bar{y}_T) - f(z) \le \frac{1}{\sum_{t=0}^{T-1} \alpha_t} \sum_{t=0}^{T-1} \alpha_t \left( f(y_{t+1}) - f(z) \right) . \tag{10}$$

Combining this with $\sum_{t=0}^{T-1} \alpha_t \ge \Omega(T^2)$, implies that in order to substantiate the proof it is sufficient to show that, $\sum_{t=0}^{T-1} \alpha_t \left( f(y_{t+1}) - f(z) \right)$, is bounded by a constant. This is the bulk of the analysis.

We start by recalling Lemma 3.1 which provides us with abound on $\alpha_t \left( f(y_{t+1}) - f(z) \right)$,

**Lemma** (Lemma 3.1). *Assume that $f$ is convex and $\beta$-smooth. Then for any sequence of non-negative weights $\{\alpha_t\}_{t \ge 0}$, and learning rates $\{\eta_t\}_{t \ge 0}$, Algorithm 2 ensures the following to hold,*

$$\alpha_t(f(y_{t+1}) - f(z)) \le (\alpha_t^2 - \alpha_t)(f(y_t) - f(y_{t+1})) + \frac{\alpha_t^2}{2}\left(\beta - \frac{1}{\eta_t}\right)\|y_{t+1} - x_{t+1}\|^2$$

$$+ \frac{1}{2\eta_t}\left(\|z_t - z\|^2 - \|z_{t+1} - z\|^2\right)$$

The proof Lemma 3.1 is provided in Appendix A.1. We are now ready to prove Theorem 3.1.

*Proof of Theorem 3.1.* According to Lemma 3.1,

$$\sum_{t=0}^{T-1} \alpha_t(f(y_{t+1}) - f(z))$$

$$\le \underbrace{\sum_{t=0}^{T-1} \frac{1}{2\eta_t}\left(\|z_t - z\|^2 - \|z_{t+1} - z\|^2\right)}_{(A)} + \underbrace{\sum_{t=0}^{T-1}(\alpha_t^2 - \alpha_t)(f(y_t) - f(y_{t+1}))}_{(B)}$$

$$+ \underbrace{\sum_{t=0}^{T-1} \frac{\alpha_t^2}{2}\left(\beta - \frac{1}{\eta_t}\right)\|y_{t+1} - x_{t+1}\|^2}_{(C)} \tag{11}$$

It is natural to separately bound each of the sums above.

**(a) Bounding** $(A)$ **:**  Using the fact that $\{1/\eta_t\}_{t \in [T]}$ is monotonically increasing we may bound $(A)$ as follows,

$$\sum_{t=0}^{T-1} \frac{1}{2\eta_t}\left(\|z_t - z\|^2 - \|z_{t+1} - z\|^2\right) \le \frac{1}{2}\sum_{t=1}^{T-1}\|z_t - z\|^2\left(\frac{1}{\eta_t} - \frac{1}{\eta_{t-1}}\right) + \frac{\|z_0 - z\|^2}{2\eta_0}$$

$$\le \frac{D^2}{2\eta_{T-1}} \tag{12}$$

where we used $\|z_t - z\| \le D$.

**(b) Bounding** $(B)$ **:**  We will require the next lemma regarding the specific choice of the weights,

**Lemma A.1.** *The following holds for the $\alpha_t$'s which are described in Eq.* (1),

$$(\alpha_t^2 - \alpha_t) - (\alpha_{t-1}^2 - \alpha_{t-1}) \le \alpha_{t-1}/2$$

Its proof appears in Appendix A.3.

We are now ready to bound (B). Recall that $z := \arg\min_{x \in \mathbb{R}^d} f(x)$, and let us denote the sub-optimality of $y_t$ by $\delta_t$, i.e. $\delta_t = f(y_t) - f(z)$. Noting that $\delta_t \geq 0$ we may show the following,

$$\sum_{t=0}^{T-1}(\alpha_t^2 - \alpha_t)\left(f(y_t) - f(y_{t+1})\right)$$

$$= \sum_{t=0}^{T-1}(\alpha_t^2 - \alpha_t)\left(\delta_t - \delta_{t+1}\right)$$

$$= \sum_{t=1}^{T-1}((\alpha_t^2 - \alpha_t) - (\alpha_{t-1}^2 - \alpha_{t-1}))\delta_t + (\alpha_0^2 - \alpha_0)\delta_0 - (\alpha_{T-1}^2 - \alpha_{T-1})\delta_T$$

$$\leq \frac{1}{2}\sum_{t=1}^{T-1}\alpha_{t-1}\delta_t$$

$$\leq \frac{1}{2}\sum_{t=1}^{T-1}\alpha_{t-1}\delta_t + \frac{1}{2}\alpha_{T-1}\delta_T$$

$$= \frac{1}{2}\sum_{t=0}^{T-1}\alpha_t\delta_{t+1}$$

$$= \frac{1}{2}\sum_{t=0}^{T-1}\alpha_t\left(f(y_{t+1}) - f(z)\right) \tag{13}$$

where in the fourth line we use Lemma A.1, we also use $\alpha_0^2 - \alpha_0 = 0$ and $\alpha_{T-1}^2 - \alpha_{T-1} \geq 0$.

**(c) Bounding** $(C)$ **:** Let us denote $\tau_\star$ as follows,

$$\tau_\star = \max\left\{t \in \{0, \ldots, T-1\} : 2\beta \geq 1/\eta_t\right\} .$$

We may now split the last term as follows,

$$\sum_{t=0}^{T-1}\frac{\alpha_t^2}{2}\left(\beta - \frac{1}{\eta_t}\right)\|y_{t+1} - x_{t+1}\|^2$$

$$= \sum_{t=0}^{\tau_\star}\frac{\alpha_t^2}{2}\left(\beta - \frac{1}{\eta_t}\right)\|y_{t+1} - x_{t+1}\|^2 + \sum_{t=\tau_\star+1}^{T-1}\frac{\alpha_t^2}{2}\left(\beta - \frac{1}{\eta_t}\right)\|y_{t+1} - x_{t+1}\|^2$$

$$\leq \frac{\beta}{2}\sum_{t=0}^{\tau_\star}\alpha_t^2\|y_{t+1} - x_{t+1}\|^2 - \frac{1}{4}\sum_{t=\tau_\star+1}^{T-1}\frac{\alpha_t^2}{\eta_t}\|y_{t+1} - x_{t+1}\|^2$$

$$= \frac{\beta}{2}\sum_{t=0}^{\tau_\star}\eta_t^2\alpha_t^2\|g_t\|^2 - \frac{1}{4}\sum_{t=\tau_\star+1}^{T-1}\eta_t\alpha_t^2\|g_t\|^2 \tag{14}$$

where in the third line we use $2\beta \leq \frac{1}{\eta_t}$ which holds for $t > \tau_\star$, implying that $\beta - \frac{1}{\eta_t} \leq -\frac{1}{2\eta_t}$; in the fourth line we use $\|y_{t+1} - x_{t+1}\| = \eta_t\|g_t\|$.

**Final Bound :** Combining the bounds in Eq. (12)-(14) into Eq. (11), we obtain,

$$\sum_{t=0}^{T-1}\alpha_t(f(y_{t+1}) - f(z)) \leq \frac{D^2}{2\eta_{T-1}} + \frac{1}{2}\sum_{t=0}^{T-1}\alpha_t\left(f(y_{t+1}) - f(z)\right)$$

$$+ \frac{\beta}{2}\sum_{t=0}^{\tau_\star}\eta_t^2\alpha_t^2\|g_t\|^2 - \frac{1}{4}\sum_{t=\tau_\star+1}^{T-1}\eta_t\alpha_t^2\|g_t\|^2$$

Re-arranging we get,

$$\frac{1}{2}\sum_{t=0}^{T-1}\alpha_t(f(y_{t+1})-f(z)) \le \underbrace{\frac{D^2}{2\eta_{T-1}} - \frac{1}{4}\sum_{t=\tau_\star+1}^{T-1}\eta_t\alpha_t^2\|g_t\|^2}_{(*)} + \underbrace{\frac{\beta}{2}\sum_{t=0}^{\tau_\star}\eta_t^2\alpha_t^2\|g_t\|^2}_{(**)} \quad (15)$$

This is the intricate part of the proof where we show that the above is bounded by a constant. This crucially depends on our choice of the learning rate, i.e., $\eta_t = 2D\left(G^2 + \sum_{\tau=0}^{t}\alpha_\tau^2\|g_\tau\|^2\right)^{-1/2}$. We require the following lemma (proof is found in Appendix A.4) before we go on,

**Lemma A.2.** *For any non-negative numbers $a_1, \ldots, a_n$ the following holds:*

$$\sqrt{\sum_{i=1}^{n}a_i} \le \sum_{i=1}^{n}\frac{a_i}{\sqrt{\sum_{j=1}^{i}a_j}} \le 2\sqrt{\sum_{i=1}^{n}a_i}\ .$$

Equipped with the above lemma and using $\eta_t$ explicitly enables to bound $(*)$,

$$
\begin{aligned}
(*) &= \frac{D}{4}\left(G^2 + \sum_{t=0}^{T-1}\alpha_t^2\|g_t\|^2\right)^{1/2} - \frac{D}{2}\sum_{t=\tau_\star+1}^{T-1}\frac{\alpha_t^2\|g_t\|^2}{\left(G^2 + \sum_{\tau=0}^{t}\alpha_\tau^2\|g_\tau\|^2\right)^{1/2}} \\
&\le \frac{D}{4}\left(\frac{G^2}{(G^2)^{1/2}} + \sum_{t=0}^{T-1}\frac{\alpha_t^2\|g_t\|^2}{\left(G^2 + \sum_{\tau=0}^{t}\alpha_\tau^2\|g_\tau\|^2\right)^{1/2}}\right) - \frac{D}{2}\sum_{t=\tau_\star+1}^{T-1}\frac{\alpha_t^2\|g_t\|^2}{\left(G^2 + \sum_{\tau=0}^{t}\alpha_\tau^2\|g_\tau\|^2\right)^{1/2}} \\
&\le \frac{DG}{4} + \frac{D}{4}\sum_{t=0}^{\tau_\star}\frac{\alpha_t^2\|g_t\|^2}{\left(G^2 + \sum_{\tau=0}^{t}\alpha_\tau^2\|g_\tau\|^2\right)^{1/2}} \\
&\le \frac{DG}{4} + \frac{D}{2}\left(\sum_{\tau=0}^{\tau_\star}\alpha_\tau^2\|g_\tau\|^2\right)^{1/2} \\
&= \frac{DG}{4} + \frac{D^2}{\eta_{\tau_\star}} \\
&\le DG/4 + 2\beta D^2 \quad (16)
\end{aligned}
$$

where in the second line we use the left hand nequality of Lemma A.2; in the fourth line we use the right hand inequality of Lemma A.2 ; and in the last line we have used the definition of $\tau_\star$ which implies that $1/\eta_{\tau_\star} \le 2\beta$.

We will also require the following lemma (proof is found in Appendix A.5),

**Lemma A.3.** *For any non-negative real numbers $a_1, \ldots, a_n$,*

$$\sum_{i=1}^{n}\frac{a_i}{1 + \sum_{j=1}^{i}a_j} \le 1 + \log\left(1 + \sum_{i=1}^{n}a_i\right)\ .$$

Equipped with the above lemma and using $\eta_t$ explicitly enables to bound $(**)$,

$$
\begin{aligned}
\frac{\beta}{2} \sum_{t=0}^{\tau_\star} \eta_t^2 \alpha_t^2 \|g_t\|^2 &= \frac{4\beta D^2}{2} \sum_{t=0}^{\tau_\star} \frac{\alpha_t^2 \|g_t\|^2}{G^2 + \sum_{\tau=0}^{t} \alpha_\tau^2 \|g_\tau\|^2} \\
&= 2\beta D^2 \sum_{t=0}^{\tau_\star} \frac{\alpha_t^2 (\|g_t\|/G)^2}{1 + \sum_{\tau=0}^{t} \alpha_\tau^2 (\|g_\tau\|/G)^2} \\
&\leq 2\beta D^2 \left( 1 + \log \left( (G/G)^2 + \sum_{\tau=0}^{\tau_\star} \alpha_\tau^2 (\|g_\tau\|/G)^2 \right) \right) \\
&= 2\beta D^2 \left( 1 + \log \left( \frac{4D^2/G^2}{\eta_{\tau_\star}^2} \right) \right) \\
&\leq 2\beta D^2 \left( 1 + 2\log \left( 4\beta D/G \right) \right) \quad (17)
\end{aligned}
$$

where in the third line we used Lemma A.3, and in the last line we have used the definition of $\tau_\star$ which implies that $1/\eta_{\tau_\star} \leq 2\beta$. Combining Equations (16), (17) back into Eq. (15) and using Jensen's inequality we are now ready to establish the final bound,

$$
\begin{aligned}
f(\bar{y}_T) - f(z) &\leq \frac{\sum_{t=0}^{T-1} \alpha_t (f(y_{t+1}) - f(z))}{\sum_{t=0}^{T-1} \alpha_t} \\
&\leq \frac{DG/2 + 8\beta D^2 \left( 1 + \log \left( 4\beta D/G \right) \right)}{T^2/32} \\
&= O\left( \frac{DG + \beta D^2 \log(\beta D/G)}{T^2} \right) .
\end{aligned}
$$

where we have used $\alpha_t \geq \frac{1}{4}(t+1)$ and therefore $\sum_{t=0}^{T-1} \alpha_t \geq T^2/32$.

$\square$

## A.1 Proof of Lemma 3.1

*Proof.* Our starting point is bounding $\alpha_t(f(x_{t+1}) - f(z))$ which can be decomposed as follows,

$$
\begin{aligned}
\alpha_t(f(x_{t+1}) - f(z)) &\leq \alpha_t g_t \cdot (x_{t+1} - z) \\
&= \alpha_t g_t \cdot (z_t - z) + \alpha_t g_t \cdot (x_{t+1} - z_t) \quad (18)
\end{aligned}
$$

where we use $g_t = \nabla f(x_{t+1})$ in conjunction with the gradient inequality. Let us now bound the terms in the above equation.

**(a) Bounding $\alpha_t g_t \cdot (z_t - z)$:** The next lemma enables to bound this term,

**Lemma A.4.** *The following holds,*

$$
\alpha_t g_t \cdot (z_t - z) \leq \left( \alpha_t g_t \cdot (z_t - z_{t+1}) - \frac{1}{2\eta_t} \|z_t - z_{t+1}\|^2 \right) + \frac{1}{2\eta_t} \left( \|z_t - z\|^2 - \|z_{t+1} - z\|^2 \right)
$$

The proof of Lemma A.4 is provided in Appendix A.2.

We can now relate the first term in the above lemma to $y_{t+1}$. Define $v = \tau_t z_{t+1} + (1 - \tau_t) y_t \in \mathcal{K}$, and notice that $x_{t+1} - v = \tau_t(z_t - z_{t+1})$. Using this we may write,

$$
\begin{aligned}
\alpha_t g_t \cdot (z_t - z_{t+1}) &- \frac{1}{2\eta_t} \|z_t - z_{t+1}\|^2 \\
&= \frac{\alpha_t}{\tau_t} g_t \cdot (x_{t+1} - v) - \frac{1}{2\eta_t \tau_t^2} \|x_{t+1} - v\|^2 \\
&= \alpha_t^2 \left( g_t \cdot (x_{t+1} - v) - \frac{1}{2\eta_t} \|x_{t+1} - v\|^2 \right) \\
&= \alpha_t^2 g_t \cdot x_{t+1} - \alpha_t^2 \left( g_t \cdot v + \frac{1}{2\eta_t} \|x_{t+1} - v\|^2 \right) \\
&\leq \alpha_t^2 g_t \cdot x_{t+1} - \alpha_t^2 \left( g_t \cdot y_{t+1} + \frac{1}{2\eta_t} \|x_{t+1} - y_{t+1}\|^2 \right) \\
&= \alpha_t^2 g_t \cdot (x_{t+1} - y_{t+1}) - \frac{\alpha_t^2}{2\eta_t} \|x_{t+1} - y_{t+1}\|^2
\end{aligned}
\tag{19}
$$

where we use $\tau_t = 1/\alpha_t$; also in the inequality we use the following equivalent form for the update rule of $y_{t+1}$,

$$
y_{t+1} = \arg\min_{x \in \mathbb{R}^d} g_t \cdot x + \frac{1}{2\eta_t} \|x - x_{t+1}\|^2 .
$$

this equivalence can be directly validated by finding the global optimum of the above objective and showing that it is obtained by choosing $y_{t+1} = x_{t+1} - \eta_t g_t$.

Combining Eq. (19) with Lemma A.4 gives,

$$
\alpha_t g_t \cdot (z_t - z) \leq \alpha_t^2 g_t \cdot (x_{t+1} - y_{t+1}) - \frac{\alpha_t^2}{2\eta_t} \|x_{t+1} - y_{t+1}\|^2 + \frac{1}{2\eta_t} \left( \|z_t - z\|^2 - \|z_{t+1} - z\|^2 \right)
\tag{20}
$$

**(b) Bounding** $\alpha_t g_t \cdot (x_{t+1} - z_t)$**:** Notice that re-arranging the relation between $x_{t+1}, y_t, z_t$ (recall $x_{t+1} = \tau_t z_t + (1 - \tau_t) y_t$) gives,

$$
x_{t+1} - z_t = r_t(y_t - x_{t+1})
\tag{21}
$$

where we denote $r_t = (1 - \tau_t)/\tau_t$. Also note that the smoothness of $f$ implies,

$$
f(y_{t+1}) - f(x_{t+1}) \leq g_t \cdot (y_{t+1} - x_{t+1}) + \frac{\beta}{2} \|y_{t+1} - x_{t+1}\|^2
\tag{22}
$$

Combining Eq. (21) and (22) we get,

$$
\begin{aligned}
g_t \cdot (x_{t+1} - z_t) \\
&= r_t \nabla f(x_{t+1}) \cdot (y_t - x_{t+1}) \\
&\leq r_t \left( f(y_t) - f(x_{t+1}) \right) \\
&= r_t \left( f(y_t) - f(y_{t+1}) \right) + (r_t + 1) \left( f(y_{t+1}) - f(x_{t+1}) \right) - \left( f(y_{t+1}) - f(x_{t+1}) \right) \\
&\leq (\alpha_t - 1) \left( f(y_t) - f(y_{t+1}) \right) + \alpha_t \left( g_t \cdot (y_{t+1} - x_{t+1}) + \frac{\beta}{2} \|y_{t+1} - x_{t+1}\|^2 \right) \\
&\quad - \left( f(y_{t+1}) - f(x_{t+1}) \right)
\end{aligned}
\tag{23}
$$

where second line uses the gradient inequality. We have also used $r_t = (1 - \tau_t)/\tau_t = \alpha_t - 1$ (see Alg. 2).

**(c) Bounding** $\alpha_t \cdot (f(y_{t+1}) - f(z))$**:**  Combining Equations (18), (20) and (23) we get,

$$\alpha_t(f(x_{t+1}) - f(z))$$
$$\leq \alpha_t g_t \cdot (z_t - z) + \alpha_t g_t \cdot (x_{t+1} - z_t)$$
$$\leq \left\{ \alpha_t^2 g_t \cdot (x_{t+1} - y_{t+1}) - \frac{\alpha_t^2}{2\eta_t}\|x_{t+1} - y_{t+1}\|^2 + \frac{1}{2\eta_t}\left(\|z_t - z\|^2 - \|z_{t+1} - z\|^2\right)\right\}$$
$$+ (\alpha_t^2 - \alpha_t)(f(y_t) - f(y_{t+1})) + \alpha_t^2 \left( g_t \cdot (y_{t+1} - x_{t+1}) + \frac{\beta}{2}\|y_{t+1} - x_{t+1}\|^2\right)$$
$$- \alpha_t (f(y_{t+1}) - f(x_{t+1}))$$
$$= (\alpha_t^2 - \alpha_t)(f(y_t) - f(y_{t+1})) + \frac{\alpha_t^2}{2}\left(\beta - \frac{1}{\eta_t}\right)\|y_{t+1} - x_{t+1}\|^2$$
$$+ \frac{1}{2\eta_t}\left(\|z_t - z\|^2 - \|z_{t+1} - z\|^2\right) - \alpha_t (f(y_{t+1}) - f(x_{t+1}))$$

Re-arranging the above equation implies,

$$\alpha_t(f(y_{t+1}) - f(z))$$
$$\leq (\alpha_t^2 - \alpha_t)(f(y_t) - f(y_{t+1})) + \frac{\alpha_t^2}{2}\left(\beta - \frac{1}{\eta_t}\right)\|y_{t+1} - x_{t+1}\|^2$$
$$+ \frac{1}{2\eta_t}\left(\|z_t - z\|^2 - \|z_{t+1} - z\|^2\right)$$

which concludes the proof.  □

## A.2  Proof of Lemma A.4

*Proof.*  Writing the update of the $z_t$'s explicitly we have,

$$z_{t+1} \leftarrow \arg\min_{x \in \mathcal{K}} \|x - (z_t - \eta_t \alpha_t g_t)\|^2 .$$

Simplifying the above implies the following equivalent form,

$$z_{t+1} \leftarrow \arg\min_{x \in \mathcal{K}} \alpha_t g_t \cdot x + \frac{1}{\eta_t}\mathcal{R}_{z_t}(x) ,$$

where $\mathcal{R}_{z_t}(x) := \|x - z_t\|^2/2$. Since $z_{t+1}$ is a solution of the above minimization problem it satisfies the first order optimality conditions, i.e. $\forall z \in \mathcal{K}$,

$$\alpha_t g_t \cdot (z - z_{t+1}) + \frac{1}{\eta_t}\nabla\mathcal{R}_{z_t}(z_{t+1}) \cdot (z - z_{t+1}) \geq 0 \tag{24}$$

which follows by the first order optimality conditions for $z_{t+1}$. We are now ready to complete the proof,

$$\alpha_t g_t \cdot (z_t - z) = \alpha_t g_t \cdot (z_t - z_{t+1}) + \alpha_t g_t \cdot (z_{t+1} - z)$$
$$\leq \alpha_t g_t \cdot (z_t - z_{t+1}) - \frac{1}{\eta_t}\nabla\mathcal{R}_{z_t}(z_{t+1}) \cdot (z_{t+1} - z)$$
$$= \alpha_t g_t \cdot (z_t - z_{t+1}) - \frac{1}{2\eta_t}\|z_t - z_{t+1}\|^2 + \frac{1}{2\eta_t}\left(\|z_t - z\|^2 - \|z_{t+1} - z\|^2\right)$$

where the second line follows due to Eq. (24), and the second line is due to following lemma (which may be easily extended to general Bergman divergences),

**Lemma A.5.** *Let* $u, v, z \in \mathbb{R}^d$, *and let* $\mathcal{R}_v(x) := \frac{1}{2}\|x - v\|^2$, *then*

$$-\nabla\mathcal{R}_v(u) \cdot (u - z) = \frac{1}{2}\|v - z\|^2 - \frac{1}{2}\|u - z\|^2 - \frac{1}{2}\|u - v\|^2$$

Below we provide the proof of this lemma.

□

### A.2.1 Proof of Lemma A.5

*Proof.* Noticing that $-\nabla\mathcal{R}_v(u) = v - u$ the lemma may be validated by a direct calculation. Indeed, $-\nabla\mathcal{R}_v(u) \cdot (u - z) = -v \cdot z + u \cdot z + u \cdot v - \|u\|^2$. Also,

$$\|v - z\|^2 - \|u - z\|^2 - \|u - v\|^2 = -2v \cdot z + 2u \cdot z + 2u \cdot v - 2\|u\|^2$$

$\square$

### A.3 Proof of Lemma A.1

*Proof.* For $t \leq 3$ we have $\alpha_t^2 - \alpha_t = 0$ and the lemma immediately follows. For $t > 3$ we have,

$$(\alpha_t^2 - \alpha_t) - (\alpha_{t-1}^2 - \alpha_{t-1}) = \frac{(t+1)^2 - 4(t+1)}{16} - \frac{t^2 - 4t}{16} = \frac{2t - 3}{8} \leq \alpha_{t-1}/2$$

$\square$

### A.4 Proof of Lemma A.2

*Proof.* **First direction:** We will prove this part by induction. The base case, $n = 1$, immediately holds. For the induction step assume that the lemma holds for $n - 1$ and let us show it holds for $n$. By the induction assumption,

$$\sum_{i=1}^{n} \frac{a_i}{\sqrt{\sum_{j=1}^{i} a_j}} \geq \sqrt{\sum_{i=1}^{n-1} a_i} + \frac{a_n}{\sqrt{\sum_{i=1}^{n} a_i}} = \sqrt{Z - x} + \frac{x}{\sqrt{Z}}$$

where we denote $x := a_n$ and $Z = \sum_{i=1}^{n} a_i$ (note that $x \leq Z$). Thus, in order to prove the lemma it is sufficient to show that,

$$\sqrt{Z - x} + \frac{x}{\sqrt{Z}} \geq \sqrt{Z},$$

which we do next. Multiplying both sides by $\sqrt{Z}$ we get that the above is equivalent to,

$$\sqrt{Z^2 - xZ} \geq Z - x$$

Taking the square of the above an re-ordering we get that the above is equivalent to,

$$x \leq Z$$

Which holds in our case since $x = a_n \leq \sum_{i=1}^{n} a_i = Z$. This concludes the first part of the proof.

**Second direction:** The second inequality in the lemma is due to Lemma 7 in [23]. For completeness we include their proof.

This part is also proved by induction. The base case, $n = 1$, immediately holds. For the induction step assume that the lemma holds for $n - 1$ and let us show it holds for $n$. By the induction assumption,

$$\sum_{i=1}^{n} \frac{a_i}{\sqrt{\sum_{j=1}^{i} a_j}} \leq 2\sqrt{\sum_{i=1}^{n-1} a_i} + \frac{a_n}{\sqrt{\sum_{i=1}^{n} a_i}} = 2\sqrt{Z - x} + \frac{x}{\sqrt{Z}}$$

where we denote $x := a_n$ and $Z = \sum_{i=1}^{n} a_i$ (note that $x \leq Z$). The derivative of the right hand side with respect to $x$ is $-\frac{1}{\sqrt{Z-x}} + \frac{1}{\sqrt{Z}}$, which is negative for $x \geq 0$. Thus, subject to the constraint $x \geq 0$, the right hand side is maximized at $x = 0$, and is therefore at most $2\sqrt{Z}$. This concludes the second part of the proof. $\square$

## A.5 Proof of Lemma A.3

*Proof.* We will prove the statement by induction over $n$. The base case $n = 1$ holds since,

$$\frac{a_1}{1 + a_1} \le 1 \le 1 + \log(1 + a_1) \,.$$

For the induction step, let us assume that the guarantee holds for $n - 1$, which implies that for any $a_1, \ldots, a_n \ge 0$,

$$\sum_{i=1}^{n} \frac{a_i}{1 + \sum_{j=1}^{i} a_j} \le 1 + \log\left(1 + \sum_{i=1}^{n-1} a_i\right) + \frac{a_n}{1 + \sum_{i=1}^{n} a_i} \,.$$

The above suggests that establishing following inequality concludes the proof,

$$1 + \log\left(1 + \sum_{i=1}^{n-1} a_i\right) + \frac{a_n}{1 + \sum_{i=1}^{n} a_i} \le 1 + \log\left(1 + \sum_{i=1}^{n} a_i\right) \,. \tag{25}$$

Using the notation $x = a_n/(1 + \sum_{i=1}^{n-1} a_i)$, Equation (25) is equivalent to the following,

$$\log(x + 1) - \frac{x}{1 + x} \ge 0 \,.$$

However, it is immediate to validate that the function $M(x) = \log(x + 1) - \frac{x}{1+x}$, is non-negative for any $x \ge 0$, which establishes the lemma. $\qquad \square$

# B Proofs for the General Convex Case (Thm. 3.2)

Here we provide the complete proof of Theorem 3.2, and of the related lemmas. For brevity, we will use $z \in \mathcal{K}$ to denote a *global minimizer* of $f$ which belongs to $\mathcal{K}$.

Recall that Algorithm 2 outputs a weighted average of the queries. Consequently, we may employ Jensen's inequality to bound its error as follow,

$$f(\bar{y}_T) - f(z) \leq \frac{1}{\sum_{t=0}^{T-1} \alpha_t} \sum_{t=0}^{T-1} \alpha_t \left( f(y_{t+1}) - f(z) \right) . \tag{26}$$

Combining this with $\sum_{t=0}^{T-1} \alpha_t \geq \Omega(T^2)$, implies that in order to substantiate the proof it is sufficient to show that, $\sum_{t=0}^{T-1} \alpha_t \left( f(y_{t+1}) - f(z) \right)$, is bounded by $\tilde{\mathcal{O}}(T^{3/2})$. This is the bulk of the analysis.

We start with the following lemma which provides us with a bound on $\alpha_t \left( f(y_{t+1}) - f(z) \right)$,

**Lemma B.1.** *Assume that $f$ is convex and $G$-Lipschitz. Then for any sequence of non-negative weights $\{\alpha_t\}_{t \geq 0}$, and learning rates $\{\eta_t\}_{t \geq 0}$, Algorithm 2 ensures the following to hold,*

$$\alpha_t(f(y_{t+1}) - f(z))$$
$$\leq \eta_t \alpha_t^2 \|g_t\|^2 + \eta_t \alpha_t^2 \|g_t\| G + \frac{1}{2\eta_t} \left( \|z_t - z\|^2 - \|z_{t+1} - z\|^2 \right) + (\alpha_t^2 - \alpha_t) \left( f(y_t) - f(y_{t+1}) \right)$$

The proof of Lemma B.1 is provided in Appendix B.1. We are now ready to prove Theorem 3.2.

*Proof of Theorem 3.2.* According to Lemma B.1,

$$\sum_{t=0}^{T-1} \alpha_t(f(y_{t+1}) - f(z))$$

$$\leq \underbrace{\sum_{t=0}^{T-1} \frac{1}{2\eta_t} \left( \|z_t - z\|^2 - \|z_{t+1} - z\|^2 \right)}_{(A)} + \underbrace{\sum_{t=0}^{T-1} (\alpha_t^2 - \alpha_t)(f(y_t) - f(y_{t+1}))}_{(B)}$$

$$+ \underbrace{\sum_{t=0}^{T-1} \eta_t \alpha_t^2 \|g_t\|^2}_{(C)} + \underbrace{\sum_{t=0}^{T-1} \eta_t \alpha_t^2 \|g_t\| G}_{(D)} \tag{27}$$

It is natural to separately bound each of the sums above.

**(a) Bounding $(A)$ :** Similarly to part $(a)$ in the proof of Theorem 3.1 we can show the following to hold,

$$\sum_{t=0}^{T-1} \frac{1}{2\eta_t} \left( \|z_t - z\|^2 - \|z_{t+1} - z\|^2 \right) \leq \frac{D^2}{\eta_{T-1}} \tag{28}$$

**(b) Bounding $(B)$ :** Similarly to part $(b)$ in the proof of Theorem 3.1 we can show the following to hold for $z = \arg\min_{z \in \mathbb{R}^d} f(x)$,

$$\sum_{t=0}^{T-1} (\alpha_t^2 - \alpha_t) \left( f(y_t) - f(y_{t+1}) \right) \leq \frac{1}{2} \sum_{t=0}^{T-1} \alpha_t \left( f(y_{t+1}) - f(z) \right) \tag{29}$$

**(c) Bounding $(C)$ :** Note that by the definition of $\eta_t$ we have

$$\eta_t = \frac{2D}{\left( G^2 + \sum_{\tau=1}^{t} \alpha_\tau^2 \|g_\tau\|^2 \right)^{1/2}} \leq \frac{2D}{\left( \sum_{\tau=1}^{t} \alpha_\tau^2 \|g_\tau\|^2 \right)^{1/2}} .$$

Using the above ineuality we get,

$$\sum_{t=0}^{T-1} \eta_t \alpha_t^2 \|g_t\|^2 \le 2D \sum_{t=0}^{T-1} \frac{\alpha_t^2 \|g_t\|^2}{\left(\sum_{\tau=0}^{t} \alpha_\tau^2 \|g_\tau\|^2\right)^{1/2}} \le 4D \sqrt{\sum_{t=0}^{T-1} \alpha_t^2 \|g_t\|^2} \qquad (30)$$

where the second inequality uses Lemma A.2.

**(d) Bounding** $(D)$ **:**   Writing down $\eta_t$ explicitly we get,

$$\sum_{t=0}^{T-1} \eta_t \alpha_t^2 \|g_t\| G = 2DG \sum_{t=0}^{T-1} \frac{\alpha_t^2 \|g_t\|}{\left(G^2 + \sum_{\tau=0}^{t} \alpha_\tau^2 \|g_\tau\|^2\right)^{1/2}}$$

$$\le 2DGT \sum_{t=0}^{T-1} \frac{\alpha_t \|g_t\|}{\left(G^2 + \sum_{\tau=0}^{t} \alpha_\tau^2 \|g_\tau\|^2\right)^{1/2}}$$

$$= 2DGT \sum_{t=0}^{T-1} \frac{\alpha_t(\|g_t\|/G)}{\left(1 + \sum_{\tau=0}^{t} \alpha_\tau^2(\|g_\tau\|/G)^2\right)^{1/2}}$$

$$\le 10DG\sqrt{\log T} \cdot T^{3/2} . \qquad (31)$$

where we used $\forall t \le T;\ \alpha_t \le T$. The last line uses the following lemma (see proof in Appendix B.2),

**Lemma B.2.** *Consider the $\alpha_t$'s used by our algorithm, i.e.,*

$$\alpha_t = \begin{cases} 1 & 0 \le t \le 2 \\ \frac{1}{4}(t+1) & t \ge 3 \end{cases}$$

*And assume a sequence of non-negative numbers, $b_0, b_1, \ldots, b_{T-1} \in [0, 1]$. Then the following holds,*

$$\sum_{t=0}^{T-1} \frac{\alpha_t b_t}{\left(1 + \sum_{\tau=0}^{t} \alpha_\tau^2 b_\tau^2\right)^{1/2}} \le 5\sqrt{\log T}\sqrt{T}$$

**Final Bound :**   Combining the bounds on the different terms, Eq. (28)-(31), together with Eq. (27), we have,

$$\sum_{t=0}^{T-1} \alpha_t(f(y_{t+1}) - f(z))$$

$$\le \frac{D^2}{\eta_{T-1}} + \frac{1}{2} \sum_{t=0}^{T-1} \alpha_t \left(f(y_{t+1}) - f(z)\right)$$

$$+ 4D\sqrt{\sum_{t=0}^{T-1} \alpha_t^2 \|g_t\|^2} + 10DG\sqrt{\log T} \cdot T^{3/2}$$

Re-arranging and using the explicit expression for $\eta_{T-1}$ we get,

$$\frac{1}{2} \sum_{t=0}^{T-1} \alpha_t(f(y_{t+1}) - f(z))$$

$$\le 5D\sqrt{G^2 + \sum_{t=0}^{T-1} \alpha_t^2 \|g_t\|^2} + 10DG\sqrt{\log T} \cdot T^{3/2}$$

$$\le 5DG\sqrt{1 + T^3} + 10DG\sqrt{\log T} \cdot T^{3/2}$$

$$\le 20DG\sqrt{\log T} \cdot T^{3/2} .$$

where we have used $\|g_t\| \leq G$, and also, $\alpha_t \leq t+1$ implying that $\sum_{t=0}^{T-1} \alpha_t^2 \leq T^3$.

Using Jensen's inequality we are now ready to establish the final bound,

$$
\begin{aligned}
f(\bar{y}_T) - f(z) &\leq \frac{\sum_{t=0}^{T-1} \alpha_t (f(y_{t+1}) - f(z))}{\sum_{t=0}^{T-1} \alpha_t} \\
&\leq \frac{40 \cdot DG\sqrt{\log T} \cdot T^{3/2}}{T^2/32} \\
&= O\left(DG\sqrt{\log T}/\sqrt{T}\right)
\end{aligned}
$$

where we have used $\alpha_t \geq \frac{1}{4}(t+1)$ and therefore $\sum_{t=0}^{T-1} \alpha_t \geq T^2/32$.

$\square$

## B.1 Proof of Lemma B.1

*Proof.* Our starting point is bounding $\alpha_t(f(x_{t+1}) - f(z))$ which can be decomposed as follows,

$$
\begin{aligned}
\alpha_t(f(x_{t+1}) - f(z)) &\leq \alpha_t g_t \cdot (x_{t+1} - z) \\
&= \alpha_t g_t \cdot (z_t - z) + \alpha_t g_t \cdot (x_{t+1} - z_t)
\end{aligned}
\tag{32}
$$

where we use $g_t = \nabla f(x_{t+1})$ in conjunction with the gradient inequality. Let us now bound the terms in the above equation.

**(a) Bounding $\alpha_t g_t \cdot (z_t - z)$:** Similarly to the proof of Lemma 3.1 we can show the following to hold (see Eq. (20) in Lemma 3.1),

$$
\alpha_t g_t \cdot (z_t - z) \leq \alpha_t^2 g_t \cdot (x_{t+1} - y_{t+1}) - \frac{\alpha_t^2}{2\eta_t}\|x_{t+1} - y_{t+1}\|^2 + \frac{1}{2\eta_t}\left(\|z_t - z\|^2 - \|z_{t+1} - z\|^2\right)
$$

Combining the above with $\|x_{t+1} - y_{t+1}\| = \eta_t\|g_t\|$ implies,

$$
\alpha_t g_t \cdot (z_t - z) \leq \eta_t \alpha_t^2\|g_t\|^2 + \frac{1}{2\eta_t}\left(\|z_t - z\|^2 - \|z_{t+1} - z\|^2\right)
\tag{33}
$$

**(b) Bounding $\alpha_t g_t \cdot (x_{t+1} - z_t)$:** Notice that re-arranging the relation between $x_{t+1}, y_t, z_t$ (recall $x_{t+1} = \tau_t z_t + (1 - \tau_t)y_t$) gives,

$$
x_{t+1} - z_t = r_t(y_t - x_{t+1})
$$

where we denote $r_t = (1 - \tau_t)/\tau_t$. Using the above we get,

$$
\begin{aligned}
g_t \cdot (x_{t+1} - z_t) \\
= r_t \nabla f(x_{t+1}) \cdot (y_t - x_{t+1}) \\
\leq (\alpha_t - 1)(f(y_t) - f(x_{t+1})) \\
\leq \alpha_t(f(y_{t+1}) - f(x_{t+1})) - (f(y_{t+1}) - f(x_{t+1})) + (\alpha_t - 1)(f(y_t) - f(y_{t+1})) \\
\leq \alpha_t G\eta_t\|g_t\| - (f(y_{t+1}) - f(x_{t+1})) + (\alpha_t - 1)(f(y_t) - f(y_{t+1}))
\end{aligned}
\tag{34}
$$

where second line uses the gradient inequality, in the third line we used $r_t = (1 - \tau_t)/\tau_t = \alpha_t - 1$ (see Alg. 2); and in the last line we used $|f(y_{t+1}) - f(x_{t+1})| \leq G\|y_{t+1} - x_{t+1}\| \leq G\eta_t\|g_t\|$, which follows by the $G$-Lipschitzness of $f$.

**(c) Bounding $\alpha_t \cdot (f(y_{t+1}) - f(z))$:** Combining Equations (32), (33), (34) we get,

$$
\begin{aligned}
\alpha_t(f(x_{t+1}) - f(z)) \\
\leq \alpha_t g_t \cdot (z_t - z) + \alpha_t g_t \cdot (x_{t+1} - z_t) \\
\leq \left\{\eta_t \alpha_t^2\|g_t\|^2 + \frac{1}{2\eta_t}\left(\|z_t - z\|^2 - \|z_{t+1} - z\|^2\right)\right\} \\
+ (\alpha_t^2 - \alpha_t)(f(y_t) - f(y_{t+1})) + \eta_t \alpha_t^2\|g_t\|G - \alpha_t(f(y_{t+1}) - f(x_{t+1}))
\end{aligned}
$$

Re-arranging the above equation and we get,

$$\alpha_t(f(y_{t+1}) - f(z))$$
$$\leq \eta_t \alpha_t^2 \|g_t\|^2 + \eta_t \alpha_t^2 \|g_t\| \|G\| + \frac{1}{2\eta_t} \left( \|z_t - z\|^2 - \|z_{t+1} - z\|^2 \right) + \left( \alpha_t^2 - \alpha_t \right) \left( f(y_t) - f(y_{t+1}) \right)$$

which concludes the proof. $\qquad\square$

## B.2 Proof of Lemma B.2

*Proof.* Let us define the following time variables,

$$T_0 = \max \left\{ t \in \{0, \ldots, T-1\} : \sum_{\tau=0}^{t} \alpha_\tau^2 b_\tau^2 \leq 1 \right\}$$

and for any $k \geq 1$

$$T_k = \max \left\{ t \in \{0, \ldots, T-1\} : 4^{k-1} < \sum_{\tau=0}^{t} \alpha_\tau^2 b_\tau^2 \leq 4^k \right\}$$

By the definition of $T_0$, the following applies,

$$\sum_{\tau=0}^{T_0} \alpha_\tau b_\tau \leq \sqrt{T_0 + 1} \left( \sum_{\tau=0}^{T_0} \alpha_\tau^2 b_\tau^2 \right)^{1/2} \leq \sqrt{T} \,. \tag{35}$$

where in the first inequality we use $\|u\|_1 \leq \sqrt{n} \|u\|_2$, $\forall u \in \mathbb{R}^n$, in the second inequality we use the definition of $T_0$ together with $T_0 \leq T - 1$.

For the other time variables we can similarly show the following bounds, i.e., $\forall k \geq 1$,

$$\sum_{\tau=T_{k-1}+1}^{T_k} \alpha_\tau b_\tau \leq \sqrt{T_k - T_{k-1}} \left( \sum_{\tau=T_{k-1}+1}^{T_k} \alpha_\tau^2 b_\tau^2 \right)^{1/2} \leq \sqrt{T_k - T_{k-1}} \cdot 2^k \tag{36}$$

where in the first inequality we use $\|u\|_1 \leq \sqrt{n} \|u\|_2$, $\forall u \in \mathbb{R}^n$, in the second inequality we use the definition of $T_k$.

Using the definition of the time variables together with Equations (35),(36) we get,

$$\sum_{t=0}^{T-1} \frac{\alpha_t b_t}{\left( 1 + \sum_{\tau=0}^{t} \alpha_\tau^2 b_\tau^2 \right)^{1/2}}$$

$$= \sum_{t=0}^{T_0} \frac{\alpha_t b_t}{\left( 1 + \sum_{\tau=0}^{t} \alpha_\tau^2 b_\tau^2 \right)^{1/2}} + \sum_{k \geq 1} \sum_{t=T_{k-1}+1}^{T_k} \frac{\alpha_t b_t}{\left( 1 + \sum_{\tau=0}^{t} \alpha_\tau^2 b_\tau^2 \right)^{1/2}}$$

$$\leq \sum_{t=0}^{T_0} \alpha_t b_t + \sum_{k \geq 1} \sum_{t=T_{k-1}+1}^{T_k} \frac{\alpha_t b_t}{\left( 1 + 4^{k-1} \right)^{1/2}}$$

$$\leq \sqrt{T} + \sum_{k \geq 1} \frac{1}{2^{k-1}} \sum_{t=T_{k-1}+1}^{T_k} \alpha_t b_t$$

$$\leq \sqrt{T} + 2 \sum_{k \geq 1} \sqrt{T_k - T_{k-1}}$$

where in the third line we use $\sum_{\tau=0}^{t} \alpha_\tau^2 b_\tau^2 > 4^{k-1}$ which by definition holds for any $T_{k-1} < t \leq T_k$.

Thus, we are left to show that $\sum_{k \geq 1} \sqrt{T_k - T_{k-1}} \leq 2\sqrt{\log T}\sqrt{T}$. To do so, first notice that the maximal value of $k$ is bounded as follows,

$$4^{k_{\max}-1} \leq \sum_{t=0}^{T-1} \alpha_t^2$$

$$\leq \sum_{t=0}^{T-1} (t+1)^2$$

$$\leq T^3$$

Thus, assuming $T \geq 2$ we have $k_{\max} \leq 3\log_2 T$, and therefore,

$$\sum_{k \geq 1} \sqrt{T_k - T_{k-1}} = \sum_{k=1}^{k_{\max}} \sqrt{T_k - T_{k-1}}$$

$$\leq \sqrt{k_{\max}} \left( \sum_{k=1}^{k_{\max}} (T_k - T_{k-1}) \right)^{1/2}$$

$$\leq \sqrt{3\log T} \, (T - T_0)^{1/2}$$

$$\leq \sqrt{3\log T}\sqrt{T} \,.$$

where we used $\|u\|_1 \leq \sqrt{n}\|u\|_2, \ \forall u \in \mathbb{R}^n$ and also $T_{k_{\max}} = T - 1$. This established the lemma. $\square$

# C  Proof of Theorem 4.1

*Proof.* For brevity we will not rehearse all of the details which are similar to the proof of the offline setting, but rather only emphasize the differences compared to the analysis of Theorem 3.2. First note the following which is analogous to Lemma B.1,

**Lemma C.1.** *Assume that $f$ is convex and $G$-Lipschitz. Assume that we invoke Algorithm 2 but provide it with noisy gradient estimates (see Eq. (9)) rather then the exact ones. Then for any sequence of non-negative weights $\{\alpha_t\}_{t\geq 0}$, and learning rates $\{\eta_t\}_{t\geq 0}$, the following holds,*

$$\alpha_t(f(y_{t+1}) - f(z))$$
$$\leq \eta_t \alpha_t^2 \|\tilde{g}_t\|^2 + \eta_t \alpha_t^2 \|\tilde{g}_t\| G + \frac{1}{2\eta_t} \left( \|z_t - z\|^2 - \|z_{t+1} - z\|^2 \right) + (\alpha_t^2 - \alpha_t)(f(y_t) - f(y_{t+1}))$$
$$+ \alpha_t(g_t - \tilde{g}_t) \cdot (z_t - z)$$

We prove this lemma in Appendix C.1.

Now, focusing on the term $\alpha_t(g_t - \tilde{g}_t) \cdot (z_t - z)$, the unbaisedness of $\tilde{g}_t$ immediately implies,

$$\mathbf{E}[\alpha_t(g_t - \tilde{g}_t) \cdot (z_t - z)] = 0 .$$

Ignoring this term and comparing the bound in the above lemma to Lemma B.1, one can see that the expression are identical up to replacing, $g_t \leftrightarrow \tilde{g}_t$. This identity in the expressions applies also to the learning rate, $\eta_t$ (again up to replacing, $g_t \leftrightarrow \tilde{g}_t$). Thus, the exact same analysis as of Lemma B.1 shows that w.p. 1 we have,

$$\sum_{t=0}^{T-1} \alpha_t(f(y_{t+1}) - f(z)) - \sum_{t=0}^{T-1} \alpha_t(g_t - \tilde{g}_t) \cdot (z_t - z) \leq O(GD\sqrt{\log T} \cdot T^{3/2}) .$$

Taking expectations and using the above in conjunction with the definition of $\bar{y}_T$ and Jensen's inequality concludes the proof. □

## C.1  Proof of Lemma C.1

*Proof.* The proof follows similar lines to the proof of Lemmas B.1 and 3.1. Here we will highlight the changes due to the stochastic setting.

Our starting point is bounding $\alpha_t(f(x_{t+1}) - f(z))$ which can be decomposed as follows,

$$\alpha_t(f(x_{t+1}) - f(z)) \leq \alpha_t g_t \cdot (x_{t+1} - z)$$
$$= \alpha_t \tilde{g}_t \cdot (z_t - z) + \alpha_t g_t \cdot (x_{t+1} - z_t) + \alpha_t(g_t - \tilde{g}_t) \cdot (z_t - z) \qquad (37)$$

Due to the unbiasedness of $\tilde{g}_t$ then the expectation of the last term $\alpha_t(g_t - \tilde{g}_t) \cdot (z_t - z)$ is zero. Let us now bound the remaining two terms in the above equation.

**(a) Bounding $\alpha_t \tilde{g}_t \cdot (z_t - z)$:**  Similarly to the proof of Lemma 3.1 we can show the following to hold (see Eq. (20) in Lemma 3.1),

$$\alpha_t \tilde{g}_t \cdot (z_t - z) \leq \alpha_t^2 \tilde{g}_t \cdot (x_{t+1} - y_{t+1}) - \frac{\alpha_t^2}{2\eta_t} \|x_{t+1} - y_{t+1}\|^2 + \frac{1}{2\eta_t} \left( \|z_t - z\|^2 - \|z_{t+1} - z\|^2 \right)$$

Combining the above with $\|x_{t+1} - y_{t+1}\| = \eta_t \|\tilde{g}_t\|$ implies,

$$\alpha_t \tilde{g}_t \cdot (z_t - z) \leq \eta_t \alpha_t^2 \|\tilde{g}_t\|^2 + \frac{1}{2\eta_t} \left( \|z_t - z\|^2 - \|z_{t+1} - z\|^2 \right) \qquad (38)$$

**(b) Bounding $\alpha_t g_t \cdot (x_{t+1} - z_t)$:**  Similarly to the proof of Lemma B.1 we can show the following to hold (see Eq. (34) therein),

$$g_t \cdot (x_{t+1} - z_t) \leq \alpha_t G \eta_t \|\tilde{g}_t\| - (f(y_{t+1}) - f(x_{t+1})) + (\alpha_t - 1)(f(y_t) - f(y_{t+1})) \qquad (39)$$

**(c) Bounding** $\alpha_t \cdot (f(y_{t+1}) - f(z))$**:** Combining Equations (38), (39) and (37) we get,

$$
\alpha_t(f(x_{t+1}) - f(z))
$$

$$
\leq \left\{ \eta_t \alpha_t^2 \|\tilde{g}_t\|^2 + \frac{1}{2\eta_t} \left( \|z_t - z\|^2 - \|z_{t+1} - z\|^2 \right) \right\} + \alpha_t(g_t - \tilde{g}_t) \cdot (z_t - z)
$$

$$
+ (\alpha_t^2 - \alpha_t) \left( f(y_t) - f(y_{t+1}) \right) + \eta_t \alpha_t^2 \|\tilde{g}_t\| G - \alpha_t \left( f(y_{t+1}) - f(x_{t+1}) \right)
$$

Re-arranging the above equation and we get,

$$
\alpha_t(f(y_{t+1}) - f(z))
$$

$$
\leq \eta_t \alpha_t^2 \|\tilde{g}_t\|^2 + \eta_t \alpha_t^2 \|\tilde{g}_t\| G + \frac{1}{2\eta_t} \left( \|z_t - z\|^2 - \|z_{t+1} - z\|^2 \right) + (\alpha_t^2 - \alpha_t) \left( f(y_t) - f(y_{t+1}) \right)
$$

$$
+ \alpha_t(g_t - \tilde{g}_t) \cdot (z_t - z)
$$

which concludes the proof.

$\square$

# D   Proof of Theorem 4.2

*Proof of Theorem 4.2.* Lets us denote by $\tilde{g}_t$ the noisy gradients received by AdaGrad upon querying $x_t$. In this case, by applying the regret guarantees of AdaGrad, [13], in conjunction to standard online to batch conversion technique, [8], implies,

$$\sum_{t=1}^{T} \mathbf{E}\left(f(x_t) - \min_{x \in \mathcal{K}} f(x)\right) \leq \mathbf{E}\sqrt{2D^2 \sum_{t=1}^{T} \|\tilde{g}_t\|^2} \tag{40}$$

Now decomposing, $\|\tilde{g}_t\| \leq \|g_t\| + \|\tilde{g}_t - g_t\|$, gives,

$$\sqrt{\sum_{t=1}^{T} \|\tilde{g}_t\|^2} \leq \sqrt{2\sum_{t=1}^{T} \|g_t\|^2 + 2\sum_{t=1}^{T} \|\tilde{g}_t - g_t\|^2} \leq \sqrt{2\sum_{t=1}^{T} \|g_t\|^2} + \sqrt{2\sum_{t=1}^{T} \|\tilde{g}_t - g_t\|^2} .$$

where the first inequality uses $(a+b)^2 \leq 2a^2 + 2b^2$, and the second inequality uses $(a+b)^{1/2} \leq a^{1/2} + b^{1/2}$ for non-negative $a, b \in \mathbb{R}$. Combining the above with Eq. (40) and applying Jensen's inequality with respect to the function $H(u) = \sqrt{u}$, gives,

$$\sum_{t=1}^{T} \mathbf{E}\left(f(x_t) - \min_{x \in \mathcal{K}} f(x)\right) \leq 2\sqrt{D^2 \sum_{t=1}^{T} \mathbf{E}\|g_t\|^2} + 2\sqrt{D^2 \sum_{t=1}^{T} \mathbf{E}\|\tilde{g}_t - g_t\|^2}$$

$$\leq 2\sqrt{2\beta D^2 \sum_{t=1}^{T} \mathbf{E}\left(f(x_t) - \min_{x \in \mathcal{K}} f(x)\right)} + 2\sqrt{\sigma^2 D^2 T} \tag{41}$$

the last line uses the lemma below, which holds since we assume $\mathcal{K}$ contains a global minimum.

**Lemma D.1.** *Let $F : \mathbb{R}^d \mapsto \mathbb{R}$ be a $\beta$-smooth function, and let $x^* = \arg\min_{x \in \mathbb{R}^d} F(x)$, then,*

$$\|\nabla F(x)\|^2 \leq 2\beta\left(F(x) - F(x^*)\right), \quad \forall x \in \mathbb{R}^d .$$

Eq. (40) enables to show, $\sum_{t=1}^{T} \mathbf{E}\left(f(x_t) - \min_{x \in \mathcal{K}} f(x)\right) \leq 4\beta D^2 + 2\sigma D\sqrt{T}$. Combining this together with the definition of $\bar{x}_T$ and Jensen's inequality concludes the proof. $\square$

## D.1   Proof of Lemma D.1

*Proof.* The $\beta$ smoothness of $F$ means the following to hold $\forall x, u \in \mathbb{R}^d$,

$$F(x + u) \leq F(x) + \nabla F(x)^\top u + \frac{\beta}{2}\|u\|^2 .$$

Taking $u = -\frac{1}{\beta}\nabla F(x)$ we get,

$$F(x + u) \leq F(x) - \frac{1}{\beta}\|\nabla F(x)\|^2 + \frac{1}{2\beta}\|\nabla F(x)\|^2 .$$

Thus:

$$\|\nabla F(x)\| \leq \sqrt{2\beta\left(F(x) - F(x + u)\right)}$$

$$\leq \sqrt{2\beta\left(F(x) - F(x^*)\right)} ,$$

where in the last inequality we used $F(x^*) \leq F(x + u)$ which holds since $x^*$ is the *global* minimum. $\square$

# E  Additional Numerical Experiments

Here, we present numerical experiments on the stochastic setting, and on a practical variant that neglects the projection steps.

## E.1  Numerical Experiments on the Stochastic Setting

We consider the same problem setup as in Section 5. Rather than using the exact gradients, we compute the unbiased estimates evaluated by a single data point (i.e. minibatch of size 1) The results are shown in Figure 2.

Figure 2: Comparison of AdaGrad and AcceleGrad in stochastic setting for smooth *(top)* and non-smooth *(bottom)* problems. Epoch denotes one full data pass, hence $500$ iterations.

AdaGrad and AcceleGrad perform similar empirically for most of the parameter choices. AdaGrad overperforms AcceleGrad only for the smooth problem with $\rho = 1$. This bahavior is caused by the projection step, and slightly increasing $D$ cures the problem for AcceleGrad.

Universal gradient methods [27] are based on a line-search technique that relies on the exact first order oracle information. Thus, it is not so surprising that in practice these methods fail upon receiving stochastic feedback, and we therefore do not present their performance.

## E.2  Numerical Experiments Neglecting the Projections

We observed that the methods work well in practice even if we ignore the projection step in the unconstrained setting. In some cases, this simple tweak may even improve the performance. We used the same test setup as in Section 5, and the results are shown in Figures 3 and 4 for the deterministic and stochastic settings respectively. Note that the method works also when we underestimate $D$.

Figure 3: Comparison of universal methods at a smooth *(top)* and a non-smooth *(bottom)* problem. Adaptive methods are tweaked to ignore the projection.

Figure 4: Comparison of AdaGrad and AcceleGrad in stochastic setting for smooth *(top)* and non-smooth *(bottom)* problems. Methods are tweaked to ignore the projection. Epoch denotes one full data pass, hence $500$ iterations.

### E.3 Experiments with Large Minibatch

In this section we apply AcceleGrad to a real world stochastic optimization problem and compare its performance with AdaGrad. We examine the effect of minibatch size verses performance. The large minibatch regime is important when one likes to apply SGD using several machines in parallel. This is done by dividing the minibatch computation between the machines. Unfortunately, it is well known that the performance of SGD degrades with the increase of minibatch size $b$. Here, we show that AcceleGrad might be more appropriate in this case.

Concretely we consider the RCV1[4] dataset which is a binary labeled set with $20424$ datapoints samples and $47366$ features. We train a classifier for this dataset using logistic loss (smooth case) as well as using the hinge loss (SVM). We compare the performance of AcceleGrad with AdaGrad.

Figure 5: Comparison of AdaGrad and AcceleGrad for logistic regression task using different minibatch sizes. We display the averaged iterates, $\bar{y}_T$ *(top)*, as well as the non-averaged iterates, $y_t$ *(bottom)*. Both methods use the same parameter $D = 10^4$.

For each method we examine several minibatch sizes, and observe the performance of each method verses the number of epochs (total number of gradients that we have computed).

The results for logistic regression appear in Figure 6. For AdaGrad we see that the performance degrades as we increase the minibatch size beyond $b = 1000$. This actually agrees with theory that predicts a degradation with the increase of $b$.

For AcceleGrad we observe an interesting phenomenon: if we aim for a very small error (in this case smaller than $10^{-2}$) then as we increase the minibatch size the performance actually improves. The intuition behind this is the following: upon using small $b$ the gradients are noisy and both AcceleGrad and AdaGrad will obtain the slow $\mathcal{O}(1/\sqrt{T})$ rate, where $T$ is the number of iterations. However, as $b$ increases the gradients are becoming more accurate and AcceleGrad with obtain a rate approaching $\mathcal{O}(1/T^2)$ while AdaGrad will approach $\mathcal{O}(1/T)$ rate. Now note that the number of gradient calculations $S$, depends on $b$ and $T$ as follows, $T = S/b$ .
Thus, for small minibatch, both methods will ensure a rate of $\mathcal{O}(\sqrt{b}/\sqrt{S})$, which clearly degrades with $b$. As $b$ increases AcceleGrad will obtain a rate approaching $\mathcal{O}(b^2/S^2)$ while AdaGrad will approach $\mathcal{O}(b/S)$ rate.

We have observed similar behaviour when train an SVM (i.e., using hinge loss). This can be seen in Figure 5.

Note that we have performed several other experiments with different $D$ parameters, and also different $\ell_3$ regularization parameters. In all experiments we have seen the same qualitative behaviour that we describe above.

Figure 6: Comparison of AdaGrad and AcceleGrad for training SVM using different minibatch sizes. We display the averaged iterates, $\bar{y}_T$ *(top)*, as well as the non-averaged iterates, $y_t$ *(bottom)*. Both methods use the same parameter $D = 10^4$.