[Reviews · NeurIPS 2018]

Reviewer 1



The universality for smooth and non-smooth problems, and ability to converge on stochastic problems are very appealing properties of the algorithm. Not having to tune the smoothness parameter seems especially attractive since it makes practical deployment of the algorithm much simpler. To the best of my knowledge, no other algorithm combines all of these properties in one package. Weaknesses: The algorithm suffers an extra log factor that would ideally be removable. The stochastic guarantee for AcceleGrad is dissapointingly non-adaptive. One would hope for an adagrad-like guarantee to complement the adagrad-like step-sizes. Clarity: The paper is clearly written. Originality: The main results seem original. I believe the stochastic result for adagrad adapting to smoothness is unofficially known among some others in the community, but there is some value in stating it formally here. Significance: As a deterministic algorithm, this algorithm seems to have significant advantages over previous universal algorithms in that it avoids line searches and does not require pre-specifying a tolerance for the final solution. The stated stochastic guarantee seems less good than the those obtained in other adaptive algorithms, but it is aesthetically appealing to have at least some guarantee. I think these kinds of adaptivity results are very interesting, and I think there is hope that the basic techniques used in the proof can form a basis for future work in the intersection of adaptivity and acceleration.

Reviewer 2



REVIEW 3218: Summary: The paper introduces a first order algorithm for unconstrained convex optimization called AcceleGrad. AcceleGrad can be applied to three different settings without any modification with ensured guarantees. It ensures O(1/T^2) for smooth objectives, O(\sqrt(log T) / \sqrt(T)) in general non-smooth objectives, and guarantees an standard rate of O(\sqrt(log T) / \sqrt(T)) in general stochastic setting where we have access to an unbiased estimation of the gradients. AcceleGrad simultaneously applies to all these three settings. The algorithm is based on an adaptive learning rate technique with importance weights together with an update that linearly couples two auxiliary sequences. Additionally the paper presents new results regarding AdaGrad algorithm in the stochastic setting with a smooth loss. The paper ensures a convergence rate of O(1/T + \sigma/\sqrt(T)) for AdaGrad in this setting. Finally the paper backs up its theoretical guarantees with experimental results. Main Comments: The paper is very well-written and straightforward to follow. The problem and the existing work are described very well. The contributions of the paper seem novel and significant in the field of convex optimization.

Reviewer 3



This paper considers the problem of designing universal gradient methods, that is designing iterative methods which given gradient or subgradient oracles automatically achieve the best rate for a given problem even without knowing all problem parameters beforehand. In particular, this paper provides a method they call AcceleGrad (which in turn is inspired from AdaGrad) which provided the initial distance of a point to the minimizer is appropriately estimated and an un upper bound on the norm of gradient is given, achieves the best rates possible for smooth minimization, non-smooth minimization, and the stochastic versions of each of these problems. Crucially, the algorithm does not need to be given the smoothness bound beforehand. The algorithm provides experiments to analyze the efficacy of the method. This paper is well-written and the proposed algorithm is very clean and some of the analysis is quite nice. This paper does a good job of unifying the analysis of multiple convex optimization problems and has key properties that as they point out, no universal method had previously. The main weaknesses of the paper are that it claims universality and optimality fairly strongly in the beginning, but requires upper bounds on the gradient and the domain size up front. While this is not necessarily a major detriment the writing should make this abundantly clear in the beginning. Also, the experiments of the paper could be strengthened. Why the particular experimental settings are chosen and what they really say about the practicality of AcceleGrad could be improved. Furthermore, it isn’t clear there is something particular novel in the analysis or how well known the open problem being addressed is. Nevertheless, this paper provides a very interesting algorithm that advances the state of the art for convex optimization and unifies the analysis of a number of techniques in a clever way. Consequently, it could make for a nice addition to the program. Specific comments: - 65: “are explore” --> “are explored” - 78: “we assume to be given” --> “we assume we are given” - 83: “Note that we allow to choose point outside K” – I would clarify this further. - 83: “We also assume that the objective function I G” – throughout the paper or just here, I would be clear. - Figure 1 – what is rho and epsilon in the charts, would make this clearer. EDIT: Thank you for your thoughtful response. Thank you for clarifying and for agreeing to make certain changes. However, I believe novelty in analysis requires more than being the first to analyze a particular algorithm, it requires an argument of why previous analysis techniques might be stuck, a simple approach might fail, etc. I still think this paper could be a nice addition to the program, but a more concrete case would be need to raise my score much more.